# Gut microbiota associations with common diseases and prescription medications in a population-based cohort

Matthew A. Jackson [1,2], Serena Verdi[1], Maria-Emanuela Maxan[3], Cheol Min Shin[1,4], Jonas Zierer [1,5], Ruth C.E. Bowyer[1], Tiphaine Martin [1,6], Frances M.K. Williams[1], Cristina Menni [1], Jordana T. Bell[1], Tim D. Spector[1] & Claire J. Steves[1,3]

The human gut microbiome has been associated with many health factors but variability between studies limits exploration of effects between them. Gut microbiota profiles are available for >2700 members of the deeply phenotyped TwinsUK cohort, providing a uniform platform for such comparisons. Here, we present gut microbiota association analyses for 38 common diseases and 51 medications within the cohort. We describe several novel associations, highlight associations common across multiple diseases, and determine which diseases and medications have the greatest association with the gut microbiota. These results provide a reference for future studies of the gut microbiome and its role in human health.

[1] Department of Twin Research and Genetic Epidemiology, King's College London, London SE1 7EH, UK. [2] Kennedy Institute of Rheumatology, University of Oxford, Oxford OX3 7FY, UK. [3] Clinical Age Research Unit, King's College Hospital Foundation Trust, London SE5 9RS, UK. [4] Department of Internal Medicine, Seoul National University Bundang Hospital, Seongnam, Gyeonggi-do, Republic of Korea. [5] Institute of Bioinformatics and Systems Biology, Helmholtz Zentrum München, 85764 Neuherberg, Germany. [6] Department of Oncological Sciences, Tisch Institute of Cancer, Icahn School of Medicine at Mount Sinai, New York, NY 10029, USA. Correspondence and requests for materials should be addressed to M.A.J. (email: matthew.jackson@kennedy.ox.ac.uk) or to C.J.S. (email: claire.j.steves@kcl.ac.uk)

The human gut microbiome has been associated with a diverse range of health deficits but there has been relatively little comparison of these effects between diseases[1]. While a recent meta-analysis found some gut microbiota associations are shared across multiple diseases[2], comparisons between studies are inherently limited by the experimental and analytical variation between them[3,4]. This can be overcome by investigating multiple phenotypes in a single well-phenotyped sample, as demonstrated by previous comparisons of the relative influence of different host factors on the gut microbiome[5,6]. A similar comparative study of human diseases requires a population with sufficient numbers of cases for multiple diseases; in this respect the British TwinsUK cohort is uniquely positioned[7]. Its members are older than other cohorts having gut microbiome data, providing a higher prevalence of common disease, and subjects have been deeply phenotyped for a range of health factors for over 25 years.

Here we describe untargeted gut microbiota association analyses with 38 common diseases within the British TwinsUK cohort. Given that medications can have a large influence on gut microbiota composition[8,9], we also explore gut microbiota associations with use of 51 common prescription medications. The results provide a reference of the relative association of different diseases and medications with gut microbiota composition at the population level.

## Results

**Gut microbiota associations with common diseases**. Disease status for individuals within the TwinsUK cohort was collated from self-reported questionnaires, and 38 diseases (those reported in >1% of the total cohort) were selected for consideration (Supplementary Data 1). Gut microbiota profiles from 16S rRNA gene sequencing of stool samples were available for 2737 individuals (89% female, age = 60 ± 12, body mass index (BMI) = 26 ± 5, mean ± SD). Within this subset, disease frequencies reflected those expected of an older female population (Fig. 1a)—the most common diseases included hypercholesterolaemia, respiratory allergies, anxiety, osteoarthritis, and hypertension; and rarer diseases included coeliac disease, epilepsy, and inflammatory bowel disease (IBD). Correlation between diseases was low with the exception of expected co-morbidities (Fig. 1b) such as within the metabolic syndrome (hypertension, hypercholesterolaemia, type 2 diabetes (T2D), and ischaemic heart disease), and between allergies, asthma, and eczema—consistent with the concept of atopy.

Microbiota data are high dimensional and inter-correlated[10]. To reduce multiple testing in association analyses we used a heuristic approach to select a minimal set of 68 taxa and diversity measures representing wider gut microbiota composition (Supplementary Data 2). Regression models were used to identify associations between the 68 microbiota markers and the 38 common diseases, adjusting for age, BMI, and technical confounders (Supplementary Data 3). Seventeen diseases had significant associations (false discovery rate (FDR) < 0.05) with at least one microbiota marker (Fig. 1c). These findings replicated reported associations including a negative association between T2D and Clostridia[11], positive associations between Enterobacteriaceae and methanogens with constipation[12], and a lower abundance of Ruminococcaceae with irritable bowel syndrome[13]. We also identified novel associations including negative associations between Prevotellaceae and food allergy; Mollicutes and Cholelithiasis; Odoribacteraceae and urinary incontinence; Deltaproteobacteria and acne; and Lentisphaeria and osteoarthritis. Among the microbiota marker traits, diversity measures had the most significant associations. Alpha diversity measures had exclusively negative associations, in accord with previous reports of reduced gut microbiome diversity in disease[1].

The power to detect associations with each disease varied with the number of cases observed. This, in combination with the additional testing from considering multiple diseases, means that associations with rarer diseases are likely under-represented. Indeed, nominally significant associations were observed with all diseases except psoriasis (Fig. 1c). These associations require validation but provide guidance for further studies to this effect. To estimate the relative scale of gut microbiota associations between diseases, we visualised the number of cases relative to the number of nominal associations observed (Fig. 1d). Conditions including IBD, T2D, constipation, recurrent urinary tract infections (UTIs), food allergies, and coeliac disease had a high number of associations despite relatively few cases, suggesting these are prime candidates for disease-specific gut microbiota studies. Conversely, few associations were observed with anxiety, respiratory allergies, and hypercholesterolaemia even with a high number of cases. We also observed diseases with few cases and few associations, such as epilepsy and gout. In these instances, the disease might either have little association with the gut microbiota or the present study is underpowered to detect associations. These results provide a reference for sample size requirements for future studies.

Age and BMI were included as covariates as they are associated with several diseases (Supplementary Fig. 1). Furthermore, as obesity associations with the gut microbiota have been examined in detail within TwinsUK we aimed to identify associations independent of these effects[14]. For comparison, we repeated the analysis without adjustment for BMI and found that obesity had the highest number of associations (Supplementary Data 4 and Supplementary Fig. 2). However, obesity was also one of the most common disorders and was correlated with several other morbidities. The results of the age- and BMI-adjusted models were also highly correlated to the results of models when adjusting for neither age nor BMI, or either one alone (Supplementary Data 4 and Supplementary Fig. 3), suggesting that these have a minimal influence on most of the disease associations observed.

**Microbiota traits with consistent associations across multiple diseases**. A recent meta-analysis by Duvallet et al.[2] showed that, together with disease-specific associations, some differences in the gut microbiota are observed across multiple diseases, which they term non-specific associations. Clustering the gut microbiota markers by their disease associations (Fig. 2), we similarly found that almost all markers had significant associations, in consistent directions, with at least two diseases. The microbiota traits could be classified into two distinct clusters that were, in general, associated with either lower or higher abundance with disease states. Several of these classifications overlap with previous studies. For example, 6 of 10 taxa identified as differentially abundant in a study of paediatric Crohn's disease patients were marker taxa in the present analysis, and all displayed consistent directions of association[15]. Conversely, Clostridiaceae and Lactobacillaceae clustered with the disease-associated microbiota traits here, but have previously been described as prevalent in healthy individuals in a review of compositional patterns observed across human gut microbiome studies[1].

As we considered marker taxa at the family and class level, our marker trait classifications could not be directly compared to the meta-analysis of Duvallet et al.[2] that defined non-specific associations at the genus level. Repeating the disease association analyses with these non-specific genera, we found reasonable clustering of genera based on their health and disease associations in the Duvallet et al.[2] study (Supplementary Fig. 4), although there were discrepancies, for instance, the genus *Veillonella* was largely at higher abundance in patients in the meta-analysis but clustered

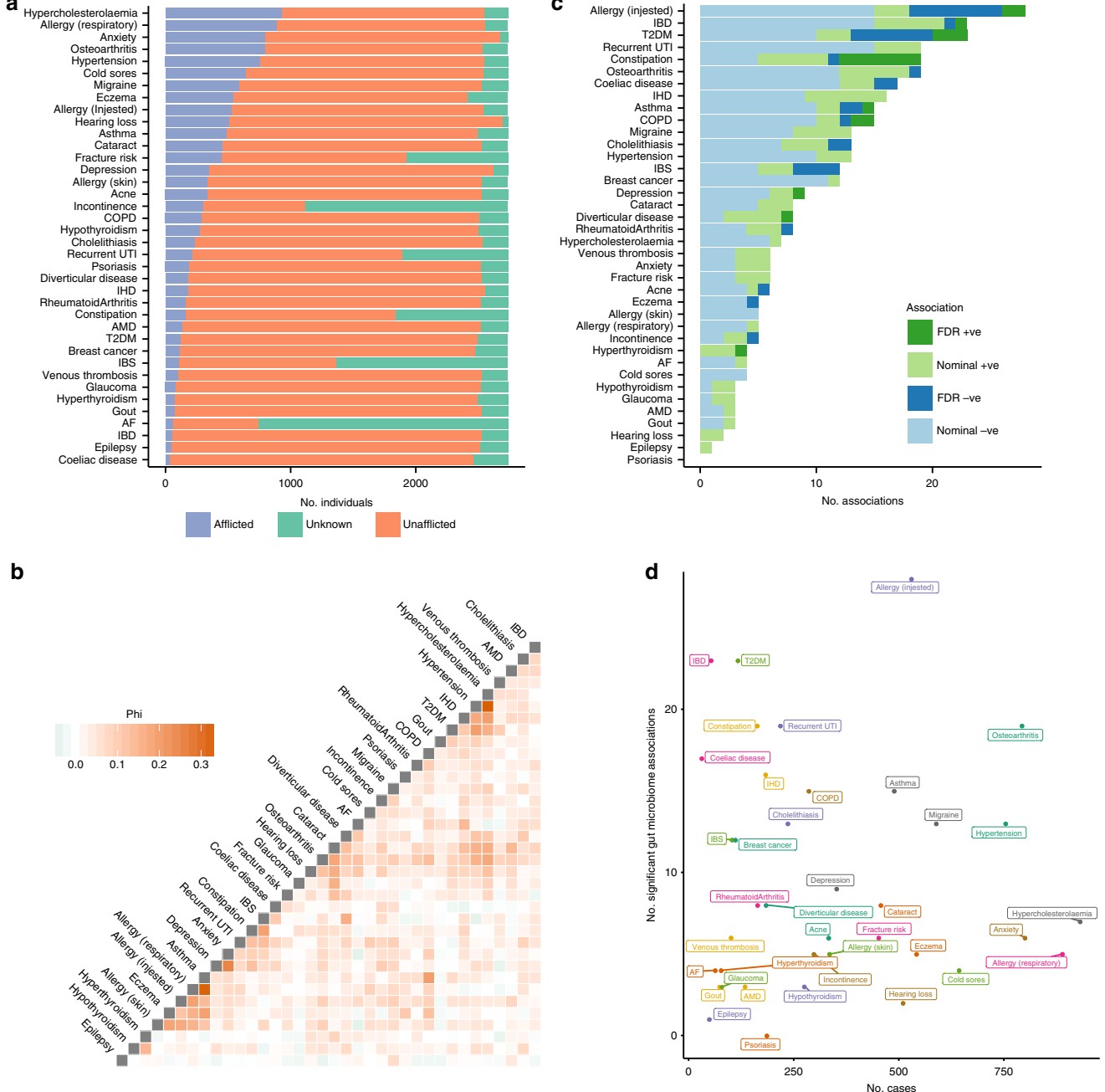

**Fig. 1** Gut microbiota associations with common diseases in TwinsUK. **a** Counts of afflicted and unafflicted individuals for common diseases within the subset of TwinsUK individuals having gut microbiota profiles. **b** Correlation between the diseases when comparing those with complete data in each pairwise comparison. Phi is equivalent to Pearson's correlation for binary variables. Breast cancer and acne are not included as they had correlation coefficients of <0.1 with all other diseases. Data overlap in each case can be found in Supplementary Data 6. **c** The number of associations observed with gut microbiota markers for each disease. Colour represents the direction of the association and darker bars represent those significant after FDR adjustment. **d** The number of afflicted individuals in the study plotted against the number of nominally significant associations observed ($p < 0.05$) for each disease

with genera generally at lower abundance with disease within the TwinsUK data. The clustering of the non-specific genera was also less distinct than observed with the class and family level marker traits. However, overall, these results contribute to increasing evidence that, at broad levels, select taxa in the gut microbiota can have consistent associations with diverse morbidities and should additionally be considered outside of a disease-specific context. Further multi-disease studies across multiple cohorts are required to identify the optimal taxa (and taxonomic levels) that define a non-

specific health-associated gut microbiota. Such taxa would be key targets for gut microbiota-based diagnostics and therapeutics and could provide insight into the mechanisms underlying gut microbiota interactions with host health.

**Gut microbiota associations with common medications.** Several studies have shown prescription medications can alter the composition of the gut microbiota[8,16–18]. These have typically

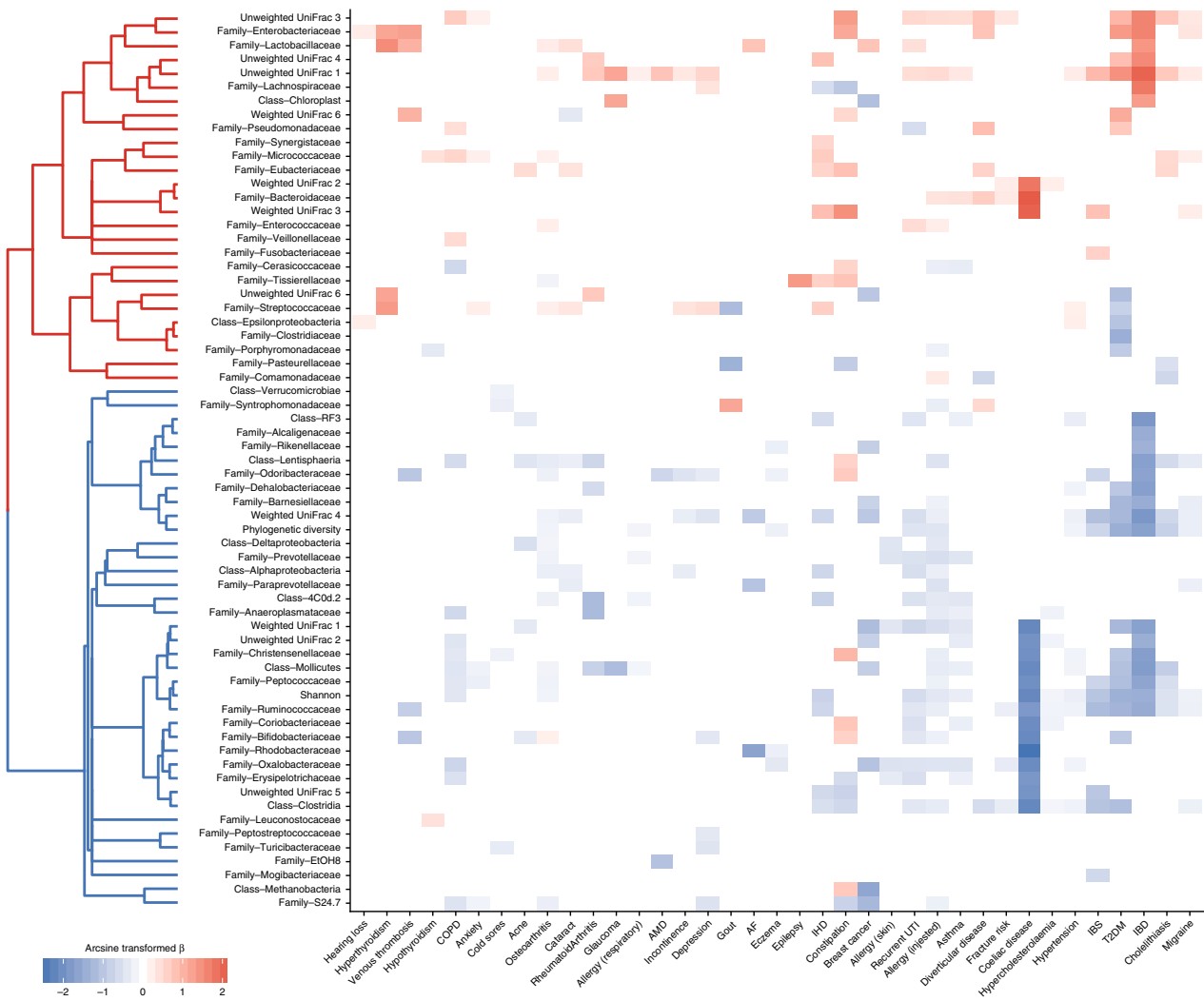

**Fig. 2** Gut microbiota traits have consistent associations with multiple diseases. Both diseases and microbiota traits have been clustered based on cosine distances generated from the beta coefficients of all nominally significant ($p < 0.05$) associations. Beta coefficients have been arcsine transformed for visualisation. Non-significant associations have been scored 0 and hence coloured white. Diseases or microbiota traits with no significant associations are not shown. Bootstrap clustering of microbiome traits identified two significant clusters highlighted in the left dendrogram; one contains traits generally at higher abundance with disease and the other traits generally at lower abundance with disease (or higher in healthy individuals)

focussed on medications expected to have a large effect, such as antibiotics[18], or those highly associated with a disease of interest, such as metformin in T2D studies[17]. There has yet to be a comprehensive investigation of associations between gut microbiota composition and the use of common medications at the level of the general population. To this end, we applied the approach used for disease comparisons to examine prescription medication use in TwinsUK.

Self-reported use of 51 prescription medications was scored from a questionnaire completed by 1724 of the individuals considered in the disease comparisons (Supplementary Data 1). Additionally, antibiotic use within the month prior to faecal sample collection was recorded separately for 2030 individuals. The most commonly used medications were statins, proton pump inhibitors (PPIs), cholecalciferol, and calcium (Fig. 3a). This reflects the age and sex of the sample and that the conditions hypercholesterolaemia and osteoarthritis were among the most prevalent. There was little correlation between the use of medications except for common known co-prescriptions such as cholecalciferol and calcium, and folic

acid and methotrexate (Fig. 3b). There was also high correlation between the usage of different inhaled medications for asthma/ COPD.

Regression models were used to identify associations between prescription medications and the gut microbiota markers as for diseases (Supplementary Data 5). Significant associations (FDR < 0.05) were observed with 19 of the 52 medications (Fig. 3c). These replicated previous observations such as higher Streptococcaceae and Micrococcaceae abundance in PPI users[8,16], and lower alpha diversity associated with both antibiotic use measures[18]. We observed several novel associations, in particular: paracetamol and opioids—both were associated with a higher abundance of Streptococcaceae and could have confounding effects in many studies given their wide usage and metabolic influences; selective serotonin reuptake inhibitors (SSRIs)—these were negatively associated with Turicibacteraceae abundance and should be explored further given the proposed association between the gut microbiota and depression[19]; and inhaled anticholinergic inhaled medications—these were negatively associated with Ruminococcaceae and Peptococcaceae abundance and alpha diversity,

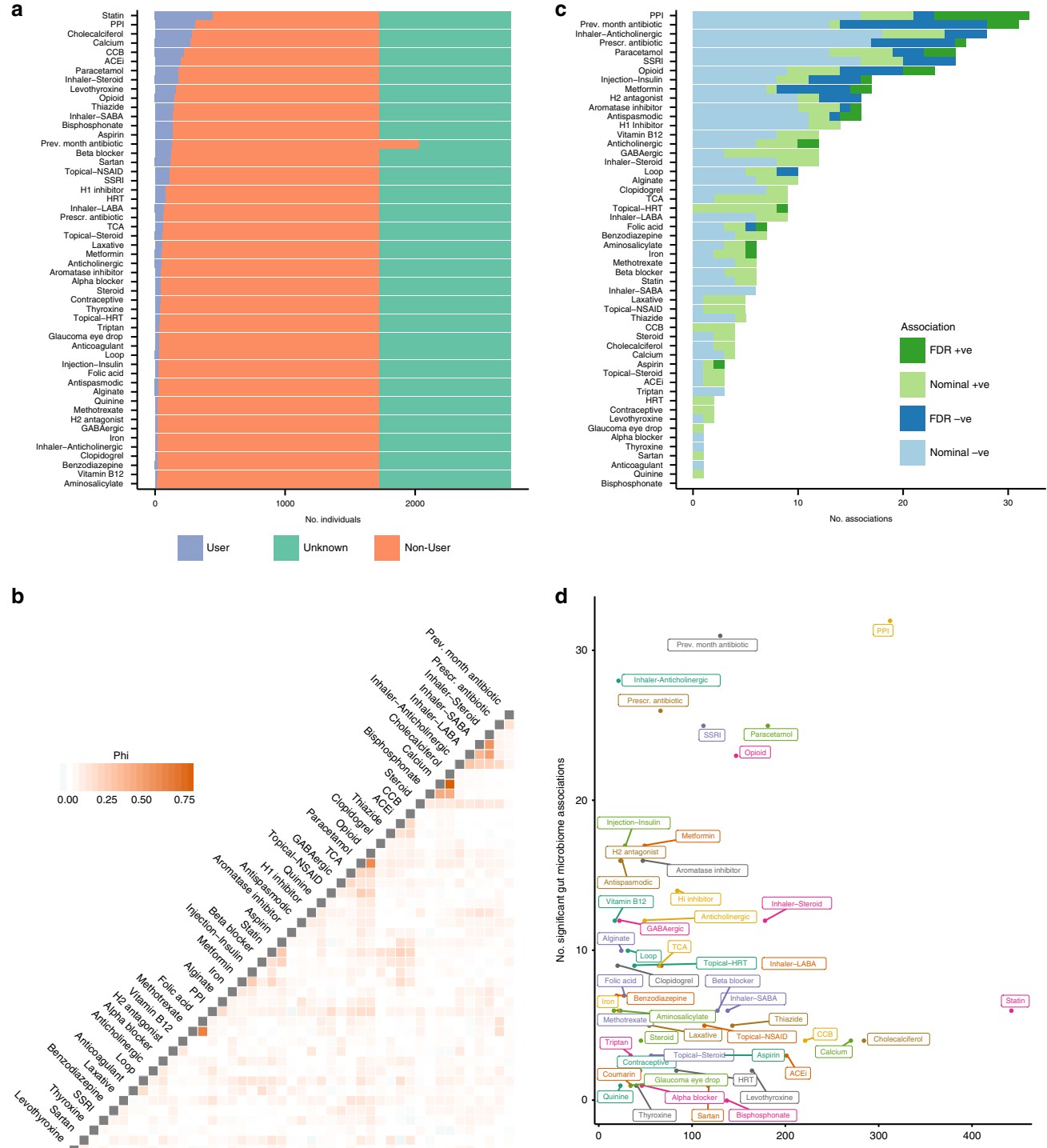

**Fig. 3** Gut microbiota associations with common prescription medications in TwinsUK. **a** Counts of users and non-users of medications within the subset of TwinsUK individuals with gut microbiota profiles. **b** Correlation between use of medications when comparing those with complete data in each pairwise comparison. Phi is equivalent to Pearson's correlation for binary variables. Medications with Phi coefficients of <0.1 with all other medications are not shown. Data overlap in each case can be found in Supplementary Data 6. **c** The number of associations observed with gut microbiota markers for each medication class. Colour represents the direction of the association and darker bars represent those significant after FDR adjustment. **d** The number of users of each medication in the study plotted against the number of nominally significant associations observed ($p < 0.05$) for each

suggesting that non-oral drug administration might indirectly influence the gut microbiota.

Similar to the disease comparisons, our power to detect associations varied by the number of medication users.

Comparing the number of nominal associations relative to the number of users of each medication, we found, reassuringly, that drugs previously associated with gut microbiota composition, notably PPIs and antibiotics, had the greatest number of

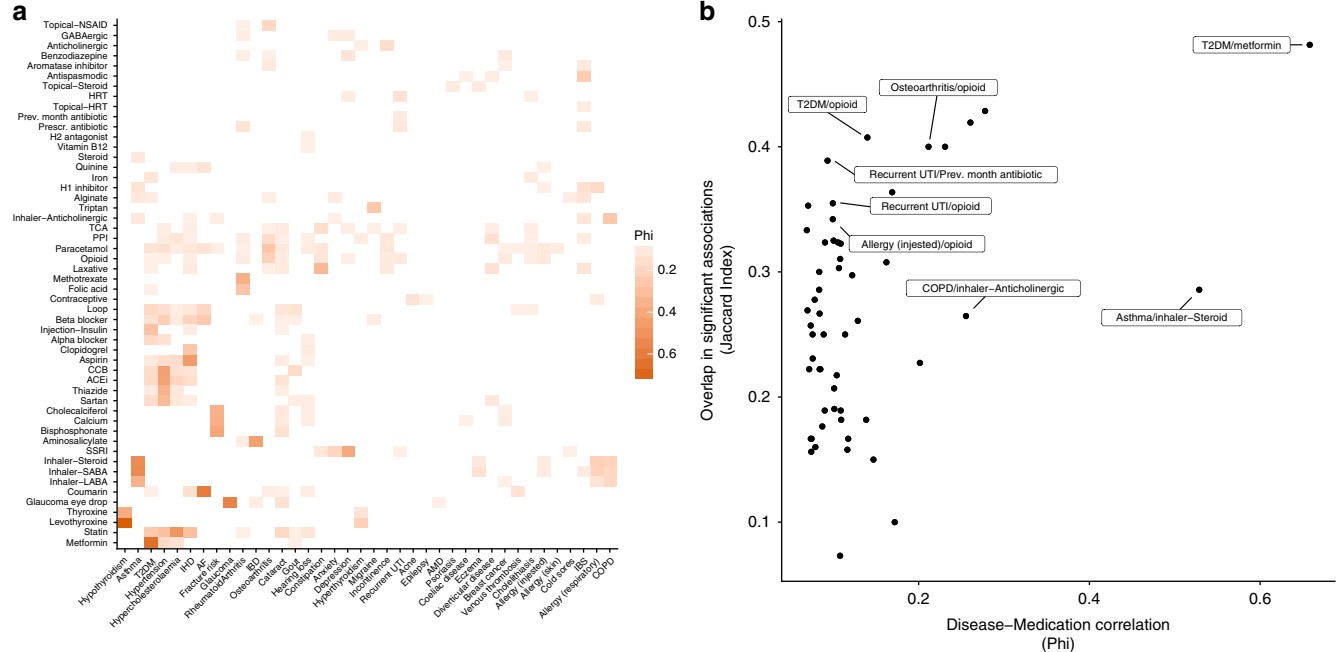

**Fig. 4** Overlap of disease and treatment associations in the gut microbiota. **a** Heatmap of the correlation between disease status and medication use status across the cohort. All non-significant correlations (FDR < 0.05) are coloured white. Rows and columns are ordered by hierarchical clustering of correlation coefficients. **b** Plot of the correlation between the significantly correlated disease–medication pairs in A versus the overlap between their associations with the gut microbiota. Showing there are cases where both correlation and overlap are high, but also those where there can be high overlap independent of correlation and vice versa. For clarity, specific examples that are discussed in the study are highlighted. A complete annotation is available in Supplementary Fig. 6

associations (Fig. 3d). Other medications having a high number of associations relative to the number of users were anticholinergic inhalers, paracetamol, SSRIs, and opioids.

Clustering microbiota traits and medications based on their associations, we observed groups of diverse medications that shared similar associations across multiple microbiota traits (Supplementary Fig. 5). This likely reflects the common microbiota associations shared across diseases. However, action of the medications on microbial abundances cannot be discounted. A recent study showed that a range of common medications have a direct influence on the growth of human gut commensals in vitro[20]. Further targeted research is warranted to examine mechanisms driving the associations with these medications and their subsequent consequences on host health. Importantly, these medications should also be considered as covariates or in screening of participants in future gut microbiome studies.

**Overlap of disease and medication associations**. There was high correlation between diseases and their associated treatments, as might be expected (Fig. 4a). For example, hypothyroidism with levothyroxine and thyroxine, T2D with metformin, and atrial fibrillation with coumarins. More widely, significant correlations were observed between numerous disease–treatment pairings, with several diseases correlating with multiple drugs and vice versa. This reflects the complex network of co-morbidities and co-prescriptions that complicates the identification of disease/medication-specific associations.

To estimate the contribution of diseases and medications to previously described observations, we explored the overlap of gut microbiota associations between correlated disease–treatment pairings (Fig. 4b and Supplementary Fig. 6). No disease–medication pairing had a complete concordance of gut microbiota associations.

Metformin and T2D had both the highest correlation and overlap in gut microbiota associations from the pairs considered, reflecting the inability to delineate effects when treatment is uniform across almost all cases. We also observed medication–disease pairs that were less correlated but had a high overlap of gut microbiota associations; these included antibiotic use and recurrent UTIs and opioids with several diseases (T2D, recurrent UTIs, food allergies, and osteoarthritis). In these instances of overlap with non-specific treatments, medication use could be responsible for a large proportion of the disease–microbiota associations. Conversely, we also observed more highly correlated disease–medication pairings that shared few gut microbiota associations; for instance, use of steroid inhalers and asthma, and anticholinergic inhaler use and chronic obstructive pulmonary disease. In these cases, separate disease and medication effects might be more prevalent. Overall, these results suggest that a complex mixture of disease- and medication-specific effects are responsible for the observed gut microbiota associations. Given the widespread use of several of the medications classes considered and the high intercorrelation of both diseases and medications, it will be important to consider nonobvious disease–medication interactions in the interpretation and design of future studies.

## Discussion

The cross-sectional and multifaceted nature of this study inherently limits our ability to delineate fully the observed associations between diseases and their associated treatments. The use of self-reported non-time-matched questionnaires for both the diseases and medications also introduces additional noise to the dataset. Hence, these results likely underestimate true effects. Further exploration of specific associations presented here will require the use of more targeted disease-specific, ideally longitudinal, studies to minimise this error and maximise the power to detect effects.

These would also provide the ability to control for other covariates that could influence both host health and the gut microbiota such as diet[21]. Intervention studies or those using treatment-naive controls will also be required to determine the specificity of associations to diseases and/or treatments. These results must also be considered within the context of a twin study. Host genetics can influence the gut microbiota and concordance rates varied across the diseases and medications considered (Supplementary Data 1)[22]. However, we expect this effect to be minimal. A recent study showed that host genetics have little influence on the gut microbiota relative to other host factors[23], and such effects would be limited to specific taxa and diseases.

Despite the limitations of the present study, we were able to identify gut microbiota associations that were applicable across multiple diseases; described novel associations with several diseases and medications; demonstrated a complex interconnectivity of morbidities, medication use, and gut microbiota associations; and described the relative association of different diseases and prescription medications with the gut microbiota at the population level. These results provide a valuable reference for future studies of the role of gut microbiota in human health.

## Methods

**Disease and medication data**. Self-reported disease data were collated from six questionnaires completed by TwinsUK participants at various times between 2002 and 2015. Most diseases were scored from the BCQ and Q11A questionnaires, which most twins had answered within 2 years of the faecal samples used to assess the gut microbiota (Supplementary Data 1 and Supplementary Fig. 7). All questions asked if a doctor or health professional had ever diagnosed the individual with the condition. Individuals were scored positive for a disease if they replied yes to any questionnaire, negative if they only replied no, and unknown if data were unavailable across all questionnaires. For constipation and cystitis, responses were scored as (0) no, (1) rarely, (2) sometimes, (3) frequently, and (4) always; in these two cases, 0–2 was considered negative and 3–4 positive. Hearing loss was classified by either doctor diagnosis, self-diagnosis, or hearing aid usage. Diseases found in at least 1% of the wider cohort were considered common and retained in analyses (Supplementary Data 1). Correlation between diseases was assessed using the Phi coefficient, the equivalent to Pearson's for binary variables.

Self-reported prescription medication use was scored from a single questionnaire. These data were cleaned to resolve spelling errors, followed by manual classification of entries into drug classes and sub-classes by a health professional. Individuals were assumed not to be taking a medication if they had completed the questionnaire without listing it. Medications used by at least 1% of the total cohort were considered for further analysis (Supplementary Data 1). Correlation between the use of different medications was determined as for diseases.

Ethics approval for the TwinsUK study was given by the NRES Committee London-Westminster (REC Reference No.: EC04/015) and all participants provided informed consent.

**Gut microbiota profiling**. This study used a larger set of gut microbiota profiles that were generated alongside those described in a recent study by Goodrich et al.[24], which reported a smaller sample as it considered only complete twin pairs. The processing of faecal samples has been described previously[22]. Briefly, samples were collected by the individual at home and either bought to a clinical visit or posted on ice to the clinical research department on ice where it was stored at −80 °C. Frozen samples were shipped to Cornell University where DNA was extracted, the V4 region of the 16S rRNA genes amplified, and amplicons sequenced using a multiplexed approach on the Illumina MiSeq platform. Sample reads were demultiplexed and paired-ends merged using a 200nt minimum overlap.

De novo chimera removal was carried out on the 16S rRNA gene sequencing per sample using UCHIME[25]. Remaining reads were collapsed to de novo operational taxonomic units (OTUs) at 97% identity using SUMACLUST within QIIME version 1.9.0[26,27]. OTU taxonomy was assigned by aligning representative sequences to the Greengenes v13_8 database using UCLUST in QIIME. Analyses were adjusted for sequencing depth throughout by using sample read count as a covariate. Taxonomic abundances were generated by collapsing OTU counts at appropriate levels, followed by conversion to log-transformed relative abundances. Three alpha diversity metrics, namely the Shannon index, phylogenetic diversity, and raw OTU counts, were calculated using QIIME. Beta diversity was calculated as both weighted and unweighted UniFrac metrics, and principal coordinate analysis of the beta distances was carried out using the vegan package[28]. The first six axes were chosen to represent beta diversity (Supplementary Fig. 8).

**Heuristic selection of microbiota marker traits**. Prior to analyses, we designed an approach to select a minimal set of microbiota marker traits for consideration. We focussed on a limited, pre-selected, set of taxonomic and diversity measures and then further reduced the redundancy of these traits based on their inter-correlation. We first restricted analyses to only consider 3 alpha diversity measures and 12 beta diversity PCoA axes, as detailed above, and all collapsed bacterial classes and families with complete taxonomic assignment. This produced an initial set of 206 gut microbiota marker traits. Spearman's correlations were calculated pairwise between these, and the correlations used to generate an adjacency matrix where correlations of >0.8 represented an edge between traits. A graphical representation of this matrix was then used for greedy selection of representative markers. Nodes (microbiota traits) were sorted by degree and the one with highest degree was then chosen as a final marker (selecting at random in the case of a tie). The marker and all connected nodes were then removed from the network and the process repeated until a final set of 68 marker traits were found such that each of the discarded traits was correlated with at least one marker.

**Disease and medication associations with gut microbiota markers**. Gut microbiota marker traits were modelled as responses in mixed effects models with technical and biological confounders including: who extracted the DNA, how the sample was collected, sequencing run, gender and family structure as random effects, and sequencing depth, age, and BMI as fixed effects. The residuals of these models were then used in disease association analyses. Individual logistic regressions were carried out with disease status as the dependent variable and residuals of microbial marker traits as independent variables. This was performed for all combinations of disease and microbiota marker traits and $p$ values were FDR adjusted to account for multiple testing using the $p$.adjust command in R. This was repeated for medication use.

Further analyses were carried out to identify disease associations using residuals that were generated without including BMI, without including age, and without including either as covariates to assess the influence of the covariates on results. We did not consider antibiotic usage as a covariate as we chose to consider it alongside the other common medications to provide an unbiased overview of disease and medication associations across the cohort.

**Clustering of microbiota marker traits by disease associations**. Beta coefficients of associations between the diseases and microbiota traits were filtered to retain only those from nominally significant associations (non-significant coefficients were considered 0). Microbiota markers and diseases without significant associations were removed. Nominal association results were used as this was a descriptive comparative analysis that did not describe association discovery (only FDR significant associations are reported as novel individual associations) and enabled clustering of the microbiota traits with less bias towards the more common diseases while providing a more conservative approach than clustering based on all beta coefficients regardless of association significance. Distance matrices between diseases and between microbiota traits were derived from the beta coefficient matrix using cosine similarity, a measure less influenced by the sparsity resulting from the zeroes of non-significant associations. Complete-linkage hierarchical clustering was used to cluster the diseases and microbiota marker traits from the cosine distance matrices using the hclust function in R, and the results visualised as a heatmap. For visualisation only, the beta coefficients were arcsine transformed to increase the visual contrast between the small coefficients and zero values. The significance of the microbiota marker clusters ($p < 0.05$) was determined by multiscale bootstrap clustering with 10,000 iterations using the pvclust package in R[29].

**Replication of non-specific genera**. Genera defined as having non-specific associations across multiple diseases (at least two) in the meta-analysis study by Duvallet et al.[2] were extracted from supplementary figure 3 of the manuscript for replication across diseases in the present study. Abundances for non-specific genera that were also observed in the TwinsUK data were adjusted for covariates including age and BMI, and the residuals used in association analyses with all diseases as previously described. Clustering of the genera and diseases and production of an associated heatmap was then carried out as for the main analyses considering all nominally significant associations.

**Correlation between disease states and medication use**. Correlation between disease states and medication use was assessed pairwise using the Phi coefficient with correlation $p$ values adjusted for multiple testing using the FDR method. Significant correlations (FDR < 0.05) were visualised as a heatmap with diseases and medications ordered by hierarchical clustering of the correlation matrix. The overlap of nominally significant ($p < 0.05$) gut microbiota associations between pairs of disease states and medications was assessed using the Jaccard index. Overlaps were compared only where diseases and medications were significantly correlated and each had at least 10 nominally significant gut microbiota associations.

**Data availability**. TwinsUK 16S rRNA gene sequencing data are available from the BioProject database under accession code PRJEB13747.

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

## Acknowledgements

The TwinsUK microbiota project was funded the National Institutes of Health (NIH) RO1 DK093595, DP2 OD007444. TwinsUK received funding from the Wellcome Trust (WT081878MA); European Community's Seventh Framework Programme (FP7/ 2007-2013), the National Institute for Health Research (NIHR)-funded BioResource, Clinical Research Facility and Biomedical Research Centre based at Guy's and St Thomas' NHS Foundation Trust in partnership with King's College London. C.J.S. was funded under a grant from the Chronic Disease Research Foundation (CDRF). T.S. is NIHR Senior investigator. C.M. is funded by the MRC AimHy (MR/M016560/1) project grant.

## Author contributions

M.A.J. and C.J.S. conceived and designed the study. M.A.J. carried out analyses. S.V., M.-E.M., C.M.S., J.Z., R.C.E.B., F.M.K.W., C.M., T.M., and C.J.S. contributed to phenotype collection and data collation. M.A.J., C.J.S., J.T.B., and T.D.S. contributed to microbiota profiling of faecal samples. M.A.J. authored the manuscript with contributions from all authors.

## Additional information

**Competing interests:** T.D.S. is co-founder of MapMySelf and MapMyGut Ltd. The remaining authors declare no competing interests.

