## [Peer Review File · Nature Communications]

Reviewers' comments:

Reviewer #1 (Remarks to the Author):

The manuscript by Jackson et al. is very straightforward - find disease and drug associations with microbiome in a single large cohort. This is by far the largest cohort this has been done in, and because protocols are identical across all diseases, it will be incredibly informative in cross-disease comparison studies to come. This is an important study.

The analyses performed are sensible, and almost certainly correct, as they confirm many of the findings of prior studies. I don't have any serious reservations about this manuscript in its current form, though I have some suggestions listed below.

MINOR QUESTIONS AND SUGGESTIONS

Feature selection:

I didn't really understand why the original set of 206 microbial features needed to be trimmed to 68. It doesn't seem like that gives much of an advantage in terms of reducing multiple hypothesis testing. And I was surprised that so many taxa were correlated at $\rho > 0.8$, that sounds really high! Are these closely-related bacterial groups?

Confounders:

I'm still a little skeptical about the correction for factors like BMI and age. First, there's no data presented that those effects are strong enough to affect the signal, except for cases where the factor is strongly confounded with disease status, like obesity. In addition, I'm not convinced that a linear correction is even appropriate.

That said, the cohort is so large, the associations could be performed for each disease using a subset of controls matched for each of these confounders. Then there'd be no need to validate the adjustment. I suspect the answers would be about the same, but to my mind the analysis would be cleaner.

Independent samples:

I didn't see any mention of the fact that the data are taken from twin pairs, and how that might affect downstream analysis. I suspect there is enough discordance between disease status, and difference in microbiome between twins that it's not a strong effect, but might be worth mentioning. In the worst case, where twins are all concordant for disease status, and carry highly similar microbiome, then p-values would have to be adjusted for fewer actual degrees of freedom, and somewhat fewer associations would be significant. Some kind of summary stat about how often twins are concordant for each disease might be useful.

Figures:

In 1b, I was unclear to what extent diseases are correlated because patient groups are overlapping?

A figure similar to 2, but showing drug associations would be very useful, perhaps for the supplemental info. The excel table is very large, and difficult to browse.

Reviewer #2 (Remarks to the Author):

The manuscript investigates how the composition of the microbe is altered by medication and co-

morbidities within a cohort of early-elderly individuals. They showed that disease is associated with a reduced diversity and altered composition within the cohort when compared to healthy individuals. They expanded on this by identifying alterations associated with medications and with disease-medication pairings. The microbial differences in the cohort appear to confirm previous association studies and add an interesting extra dimension in the form of the correlation between microbiome changes and the use of medications.

The authors point out that previous studies that have provided meta-analyses from multiple sources are inherently flawed due to the use of different experimental and analytical techniques. They counter these problems using a large in-house dataset with an in-depth analysis approach. Whilst the authors should be commended on the analysis approach they took in such a complicated dataset. The fact that is a challenging dataset means that there are a number of statistical points that need to be clarified before publication.

General comments:

1. Please provide columns showing the number of cases per disease/medication to supplementary table 1.
2. The use of nominally significant results needs to be justified in the main text.
3. Why were logistic models with the disease as the response variable used? Why not use linear and negative binomial models that have the disease state as a predictor and the microbiome variable under investigation as a response and in this way control for confounders when testing for association to medication and disease state?
4. It is unusual for a study to include antibiotic treated samples within the cohort. Is there any adjustment for antibiotic usage in the models?
5. Microbiota variables were adjusted for BMI and age. Having age and BMI in the model by default is may not be optimal as many diseases are not BMI or age-related and if there is an adjustment for age for a microbiome taxon, then that may lead to false negative or false positive results. Therefore, this adjustment may not be appropriate in all cases. The authors removed BMI in a repeated analysis and reported the result. They should do the same for age and present the comparison of the results with and without BMI and age within a supplementary document.
6. "We also observed diseases with few cases and few associations, such as epilepsy and gout. In these instances, this study may be underpowered to detect associations and these results provide a valuable reference for sample size requirements for future studies." This line assumes that there will be associations in a larger study. This may not be the case. The line should be reworded to reflect this possibility.
7. The authors state that "as we considered marker taxa at the family and class level, our classifications could not be directly compared to the aforementioned meta-analysis that defined non-specific associations at the genus level". This can be easily remedied by testing the genus level associations and confirming if these genera are differentially abundant and therefore, may be responsible or partially responsible for the observations in this study

Reviewer #1 (Remarks to the Author):

The manuscript by Jackson et al. is very straightforward - find disease and drug associations with microbiome in a single large cohort. This is by far the largest cohort this has been done in, and because protocols are identical across all diseases, it will be incredibly informative in cross-disease comparison studies to come. This is an important study.

The analyses performed are sensible, and almost certainly correct, as they confirm many of the findings of prior studies. I don't have any serious reservations about this manuscript in its current form, though I have some suggestions listed below.

MINOR QUESTIONS AND SUGGESTIONS

Feature selection:

I didn't really understand why the original set of 206 microbial features needed to be trimmed to 68. It doesn't seem like that gives much of an advantage in terms of reducing multiple hypothesis testing. And I was surprised that so many taxa were correlated at $\rho > 0.8$, that sounds really high! Are these closely-related bacterial groups?

We designed the approach to reduce the number of traits from the entire microbiome data set a priori, so did not know how many family and class level traits we would have before the further dimensionality reduction. However, the minimal trait set still reduced the tests three-fold compared to just selecting the complete family and class assignments. We have now highlighted within the manuscript that the approach was designed prior to any analyses were carried out.

The high correlations were indeed often between closely related taxonomic groups (the marker traits that represented each of the original 206 traits are shown in Supplementary Table 2). This is in part due to similar properties being shared by more closely related taxa, and our consideration of taxa at the family and class level. The classes which contained only one or few bacterial families will not have significantly different abundances to their child families. There was also high correlation between the various indices of microbiome diversity. The high level of inter-correlation was expected and was the main motivation for our dimensionality reduction approach to minimise the redundancy in our testing.

Confounders:

I'm still a little skeptical about the correction for factors like BMI and age. First, there's no data presented that those effects are strong enough to affect the signal, except for cases where the factor is strongly confounded with disease status, like obesity. In addition, I'm not convinced that a linear correction is even appropriate.

That said, the cohort is so large, the associations could be performed for each disease using a subset of controls matched for each of these confounders. Then there'd be no need to validate the adjustment. I suspect the answers would be about the same, but to my mind the analysis would be cleaner.

We have improved the discussion of age and BMI adjustment in this version of the manuscript. Rather than using matched controls, we have addressed this by repeating the analyses with disorders for all possible combinations of age, BMI, and neither as covariates (as suggested by reviewer 2, point 5). There is extremely high correlation between the beta coefficients and p-values in each case suggesting that age and BMI are not having large effects in described associations. These results are presented in additional tables in Supplementary Table 4 and in Supplementary Figure 3.

Independent samples:

I didn't see any mention of the fact that the data are taken from twin pairs, and how that might affect downstream analysis. I suspect there is enough discordance between disease status, and difference in microbiome between twins that it's not a strong effect, but might be worth mentioning. In the worst case, where twins are all concordant for disease status, and carry highly similar microbiome, then p-values would have to be adjusted for fewer actual degrees of freedom, and somewhat fewer associations would be significant. Some kind of summary stat about how often twins are concordant for each disease might be useful.

Thank you for highlighting this, it is an important consideration that should be discussed. We have now included a discussion of this effect in the manuscript, including a reference to a recent study showing genetic effects have a small impact on the gut microbiome relative to other factors. We have also now expanded Supplementary Table 1 to show the disease concordance and discordance rates for the monozygotic and dizygotic twin pairs with complete data.

Figures:

In 1b, I was unclear to what extent diseases are correlated because patient groups are overlapping?

We have since tried to include overlap size in the plot but it was unreadable, so we now include a table of the overlap size, this also has the correlation coefficients in text format. This has also been done for the medications as in figure 3b. These can be found in the new Supplementary Table 6.

A figure similar to 2, but showing drug associations would be very useful, perhaps for the supplemental info. The excel table is very large, and difficult to browse.

We have generated an identical plot but for the drug associations. This is included in Supplementary Figure 5 and is discussed within the manuscript.

Reviewer #2 (Remarks to the Author):

The manuscript investigates how the composition of the microbiome is altered by medication and comorbidities within a cohort of early-elderly individuals. They showed that disease is associated with a reduced diversity and altered composition within the cohort when compared to healthy individuals. They expanded on this by identifying alterations associated with medications and with disease-medication pairings. The microbial differences in the cohort appear to confirm previous association studies and add an interesting extra dimension in the form of the correlation between microbiome changes and the use of medications.

The authors point out that previous studies that have provided meta-analyses from multiple sources are inherently flawed due to the use of different experimental and analytical techniques. They counter these problems using a large in-house dataset with an in-depth analysis approach. Whilst the authors should be commended on the analysis approach they took in such a complicated dataset. The fact that is a challenging dataset means that there are a number of statistical points that need to be clarified before publication.

General comments:

1. Please provide columns showing the number of cases per disease/medication to supplementary table 1.

These, alongside the concordance metrics noted by Reviewer 1, have now been added to Supplementary Table 1.

2. The use of nominally significant results needs to be justified in the main text.

We have now justified the use of the nominal association results in more detail in the methods where appropriate. We have also added an additional note to the main text that explicitly states that the nominal associations require further testing and should be considered as a guide for future studies.

3. Why were logistic models with the disease as the response variable used? Why not use linear and negative binomial models that have the disease state as a predictor and the microbiome variable under investigation as a response and in this way control for confounders when testing for association to medication and disease state?

In previous studies^{1,2}, we have used the suggested approach (modeling the microbiome trait as the dependent variable) and have found it returned almost identical results to the approach used here. Here, we chose to carry out separate adjustment for the microbiome trait variance and then use the residuals of these traits in the models principally for two reasons. Firstly, the technical covariate data, age and BMI were all collected at the same time point as the microbiome sample whereas the disease and medication data were collated from questionnaires taken across the last 10 years. Secondly, we found that this approach was less computationally intensive and provided an efficient and uniform method to carry out the association analyses across the high number of diseases and medications considered whilst producing almost identical results. We are confident in the results produced by this method given the consistency of the results in the new analyses without BMI and age as covariates (see response to point 5), and the broad agreement between this manuscript and results from our previous studies (which have used the alternate modeling approach) and within our cohort and others.

4. It is unusual for a study to include antibiotic treated samples within the cohort. Is there any adjustment for antibiotic usage in the models?

No we did not adjust for it at any point. As the basis of this study was untargeted description of the associations observed with the diseases and medications at the population level, we chose to treat antibiotics in the same manner as all other common medications and not consider the influences on one another until the correlative analysis towards the end of the manuscript. This enables us to present the results as disease associations not adjusting for medication (and vice versa) for all diseases and medications without biasing towards previously established microbiome-medication associations. We have now made this clearer by adding an explicit statement to this effect to the methods section.

5. Microbiota variables were adjusted for BMI and age. Having age and BMI in the model by default is may not be optimal as many diseases are not BMI or age-related and if there is an adjustment for age for a microbiome taxon, then that may lead to false negative or false positive results. Therefore, this adjustment may not be appropriate in all cases. The authors removed BMI in a repeated analysis and reported the result. They should do the same for age and present the comparison of the results with and without BMI and age within a supplementary document.

Thank you, we agree that this would be a great improvement to the manuscript. We have since repeated the analyses again with adjustment for BMI alone, and neither age nor BMI. These results are now presented in Supplementary Table 4, we also provide an additional supplement (Supplementary Figure 6) visualising the correlation between the results of the models in each case, which are also discussed within the manuscript results.

6. “We also observed diseases with few cases and few associations, such as epilepsy and gout. In these instances, this study may be underpowered to detect associations and these results provide a valuable reference for sample size requirements for future studies.” This line assumes that there will be associations in a larger study. This may not be the case. The line should be reworded to reflect this possibility.

This line has been edited to better reflect that it could also be the case that these diseases have little association with the gut microbiota.

7. The authors state that “as we considered marker taxa at the family and class level, our classifications could not be directly compared to the aforementioned meta-analysis that defined non-specific associations at the genus level”. This can be easily remedied by testing the genus level associations and confirming if these genera are differentially abundant and therefore, may be responsible or partially responsible for the observations in this study

We did not test the genus level associations as we wanted to maintain the testing to the limited set of marker traits we chose prior to all analyses, but do agree that this simple test could be of much value. However, we have limited these new analyses to just replicating the reported genera from the meta-analysis study. This is to avoid adding an additional table of all potential genus level associations that are not appropriately considered in our existing false-discovery rate corrections.

We found that the disease associations largely clustered the non-specific genera by the disease classifications defined in the meta-analysis paper. However, there were exceptions and the clusters were not as clearly defined as those observed in our analyses at the family and class level; reaffirming the need for further studies to identify an optimal approach to identify non-specific microbiome markers of health/disease. This analysis is now discussed in the manuscript and is accompanied by Supplementary Figure 4 showing the clustering of the non-specific genera by their disease associations in the TwinsUK data.

References

1. Jackson, M. A. *et al.* Proton pump inhibitors alter the composition of the gut microbiota. *Gut* **65**, 749–756 (2016).
2. Jackson, M. A. *et al.* Signatures of early frailty in the gut microbiota. *Genome Med.* **8**, 8 (2016).

REVIEWERS' COMMENTS:

Reviewer #2 (Remarks to the Author):

After reviewing the authors responses I have no further major remarks and am happy for this paper to be published after two minor corrections.

Minor

On line 78 to 79, some punctuation or rewriting is necessary as as written it sounds like the authors are saying that diversity is negatively associated with diversity.

lines 120 to 121, I think this statement should have a slightly expanded explanation.

I do not need to review this manuscript again.

Responses to reviewer comments

On line 78 to 79, some punctuation or rewriting is necessary as as written it sounds like the authors are saying that diversity is negatively associated with diversity.

The sentence has now been split to clarify the meaning. The previous and new versions are copied below.

Previous:

“Amongst the microbiota marker traits, diversity measures had the most significant associations with alpha diversity measures having exclusively negative associations, in accord with previous reports of reduced gut microbiome diversity in disease¹.”

New:

” Amongst the microbiota marker traits, diversity measures had the most significant associations. Alpha diversity measures had exclusively negative associations, in accord with previous reports of reduced gut microbiome diversity in disease¹.”

lines 120 to 121, I think this statement should have a slightly expanded explanation.

This has also now been clarified. See the changes below.

Previous:

“Conversely, several of the marker taxa considered were classified in opposing directions in relation to a recent review summarising trends across several disease-specific studies¹. ”

New:

“Conversely, Clostridiaceae and Lactobacillaceae clustered with the disease-associated microbiota traits here, but have previously been described as prevalent in healthy individuals in a review of compositional patterns observed across human gut microbiome studies¹.”